# AlphaTablets: A Generic Plane Representation for 3D Planar Reconstruction from Monocular Videos

**Yuze He**[1], **Wang Zhao**[1], **Shaohui Liu**[2], **Yubin Hu**[1],
**Yushi Bai**[1], **Yu-Hui Wen**[3], **Yong-Jin Liu**[1]*
[1]Tsinghua University    [2] ETH Zurich    [3] Beijing Jiaotong University

## Abstract

We introduce AlphaTablets, a novel and generic representation of 3D planes that features continuous 3D surface and precise boundary delineation. By representing 3D planes as rectangles with alpha channels, AlphaTablets combine the advantages of current 2D and 3D plane representations, enabling accurate, consistent and flexible modeling of 3D planes. We derive differentiable rasterization on top of AlphaTablets to efficiently render 3D planes into images, and propose a novel bottom-up pipeline for 3D planar reconstruction from monocular videos. Starting with 2D superpixels and geometric cues from pre-trained models, we initialize 3D planes as AlphaTablets and optimize them via differentiable rendering. An effective merging scheme is introduced to facilitate the growth and refinement of AlphaTablets. Through iterative optimization and merging, we reconstruct complete and accurate 3D planes with solid surfaces and clear boundaries. Extensive experiments on the ScanNet dataset demonstrate state-of-the-art performance in 3D planar reconstruction, underscoring the great potential of AlphaTablets as a generic 3D plane representation for various applications. Project page is available at: `https://hyzcluster.github.io/alphatablets`.

## 1 Introduction

3D planar reconstruction from monocular videos is a crucial aspect of 3D computer vision, dedicated to the precise detection and reconstruction of underlying 3D planes from consecutive 2D imagery. The reconstructed 3D planes serve as a flexible representation of surfaces, facilitating various applications such as scene modeling, mixed reality and robotics.

Traditional methods for 3D planar reconstruction rely heavily on explicit geometric inputs [36, 41], hand-crafted features [6, 4], strong assumptions [10, 15] and solvers [37, 42, 16] to detect and reconstruct the planes, thereby imposing limitations on scalability and robustness. Learning-based methods [27, 26, 2, 28, 48] leverage the power of data-driven training to directly segment the plane instances and regress the plane parameters from single or sparse-view images. Notably, PlanarRecon [50], a pioneering monocular video-based learning method, operates within the 3D volume space to detect and track planes, and has demonstrated promising results in recovering planar structures. However, it often falls short in detecting complete planar reconstructions and struggles with generalization across diverse scenes. How to build an accurate, complete and generalizable 3d planar reconstruction system is still a challenging open problem.

We inspect this problem from the perspective of plane representation. Current methodologies employ various representations, such as view-based 2D masks [42, 16, 3, 9, 26, 2, 48], 3D points [37, 30], surfels [36], and voxels [50]. While 2D masks can precisely illustrate plane contours, this 2D plane representation faces inconsistencies across different views and necessitates complex matching and fusion processes to reconstruct 3D surfaces. In contrast, 3D representations directly depict 3D planar surfaces. However, they suffer from discontinuous geometry and texture due to discretized sampling, and struggle to accurately model complex plane boundaries.

---

*Corresponding Author.

38th Conference on Neural Information Processing Systems (NeurIPS 2024).

To address above limitations, we propose a novel plane representation termed AlphaTablets. AlphaTablets define 3D planes as rectangles with alpha channels, providing a natural delineation of irregular plane boundaries and enabling continuous solid 3D surface representation. AlphaTablets combine the best of 2D and 3D plane presentations: by defining and optimizing in 3D, AlphaTablets ensure efficiency and consistency across all views; meanwhile, by introducing alpha channels and texture maps in plane's canonical coordinates, AlphaTablets can accurately model solid surfaces and complex irregular plane boundaries. Furthermore, we introduce rasterization formulations for AlphaTablets, facilitating differentiable rendering into images.

Based on AlphaTablets, we present a bottom-up pipeline for 3D planar reconstruction from posed monocular videos. Initially, leveraging 2D superpixels [1], we initialize the AlphaTablets using pre-trained depth and surface normal models [19, 12]. This initialization yields a dense yet noisy assembly of relatively small and overlapping AlphaTablets. Next, these small planes are optimized via differentiable rendering with hybrid regularizers to adjust both geometry, texture and alpha channels. We further introduce an effective merging scheme that facilitates the fusion of neighboring tablets, thereby promoting the growth of larger, cohesive planes. Through iterative cycles of optimization and merging, our final reconstructions boast solid surfaces, clear boundaries, and interpolatable texture maps, delivering accurate 3D planar structures. Moreover, our approach enables flexible plane-based scene editing. Extensive experiments on ScanNet [11] dataset demonstrate state-of-the-art performance of 3D planar reconstruction, showing the great potential of being a generic 3D plane representation for subsequent applications.

In summary, our contributions are threefold:

- We propose AlphaTablets, a novel and generic 3D plane representation which features effective and flexible modeling of plane geometry, texture and boundaries. We derive the rasterization formulation for AlphaTablets, enabling differentiable rendering to images.

- We build an accurate and generalizable 3D planar reconstruction system from monocular videos upon AlphaTablets. The key components are effective initialization from pre-trained monocular cues, differentiable rendering based optimization, and the proposed merging mechanism for AlphaTablets.

- The proposed system achieves state-of-the-art performance for 3d planar reconstruction, while also enabling flexible plane-based scene editing for various applications.

## 2 Related Works

**Classical 3D Plane Reconstruction.** Traditional 3D plane reconstruction methods typically fit planes from geometric inputs like RGB-D images [36, 41, 35, 21] and 3D point cloud [7, 43], using robust estimators such as RANSAC [14] and its variants [37]. Another line of research approaches utilize multi-view images as input. Early works [42, 16, 15, 3] detect 3D planes from reconstructed sparse 3D points and lines, and then optimize plane masks as multi-label segmentation through image-based solvers such as Markov Random Fields (MRF) and Graph Cut [23]. Argiles et al. [3] propose a plane consistency check to determine plane boundaries using square cells rather than sparse point contours. Manhattan world assumption is introduced [10, 53] to better reconstruct the dominant planes. While these methods can segment 2D complex planes, the 2D mask representation of planes causes view inconsistency, visibility issues and requires additional efforts to reconstruct 3D planes. In contrast, AlphaTablets directly model and optimize planes in 3D with differentiable rendering, eliminating the inconsistency and occlusion problems. Given a monocular video as input, DPPTAM [9] reconstructs both the 3D planar and non-planar structure in a SLAM fashion, and adapt superpixel to group homogeneous pixels for textureless regions. Our method also uses superpixels as 2D units to initialize AlphaTablets, but provides greater flexibility in adjusting geometry, texture, and boundaries through optimization.

**Learning-based 3D Plane Reconstruction.** Data-driven methods leverage large-scale training data to learn geometric priors, enabling the reconstruction of 3D planes from single or sparse view images. PlaneNet [27], PlaneRecover [52] and PlaneRCNN [26] create training datasets for plane detection and reconstruction, and train deep neural networks to directly segment plane instances and regress plane parameters from single RGB image. Further enhancements along this line include the use of transformer architectures [47], multi-task collaboration [40], pairwise plane relations [34], and feature clustering [54], etc. Due to the highly ill-posed nature of single-image plane reconstruction, many

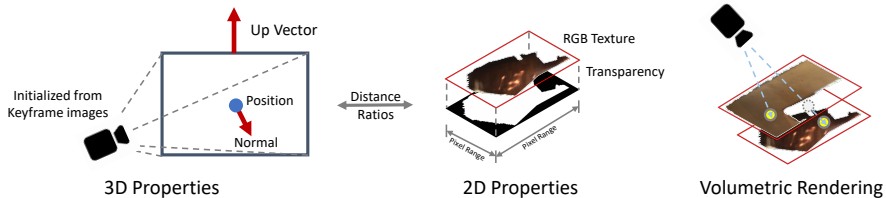

Figure 1: **Illustration of tablet properties and rendering.** Normal and up vector determines the rotation of a tablet in 3D space, while every tablet maintains a distance ratio between the coordinates of the 3D field and 2D-pixel space.

works [2, 22, 28, 48] adopt sparse view images as inputs, and explore joint plane detection, association and optimization to help the final reconstruction. However, these methods can only recover local 3D planar structures from sparse views, and it is challenging to extend them to image sequences. PlanarRecon [50] proposes a 3D volume-based planar reconstruction system with plane detection and tracking modules that sequentially process video frames and update 3D planar reconstructions. While effective in producing clean and consistent 3D planes, PlanarRecon often oversimplifies planar structures, resulting in incomplete reconstructions. Furthermore, being trained on indoor datasets with gravity-aligned camera poses, PlanarRecon struggles to generalize to unseen data. On the contrary, our method optimizes AlphaTablets for each scene, thus can generalize to any video sequence.

**3D Planar Surface Representation.** While most of current works employ 2D plane representation for detection and reconstruction, 3D representation is more efficient and consistent by aggregating 2D image information directly in 3D. 3D point cloud [37, 38, 33, 51, 32] is one of the most widely used 3D surface representation, yet they are discrete samples of a surface and cannot fully capture continuous geometry and textures. Extending points into planar primitives, surfels [36, 17] and 2D gaussians [18, 20] represent surface with 2D disks in 3D space, and demonstrate impressive results for both planar and non-planar surfaces. Unfortunately, both surfels and 2D gaussians only provide local planar structure within one unit, making it difficult to directly represent large geometric planes.

In terms of continuous surface representation, mesh is the most popular representation with mature graphic pipelines. Many works [50, 42, 49, 39] convert other 3D surface representations into mesh for visualization or further optimization. While meshes using 3D triangles or rectangles can represent 3D planes with regular quadrangular shapes, they struggle with irregular complex boundaries and are difficult to build and optimize from scratch. AlphaTablets inherit the advantages of continuous geometry and canonical texture modeling from mesh, and further introduce alpha channels to handle the irregular plane boundaries, acting as "rectangle soup with alpha channels". With the popularity of implicit neural representation, several works [25, 8] have explored encoding 3D plane primitives with MLPs. Compared to implicit representations, AlphaTablets offer the advantages of explicit representation, such as easy editing and fast rendering without network inference.

## 3 Method

Our proposed AlphaTablets is a novel and generic 3D plane representation, which enables accurate and generalizable 3D planar reconstruction from monocular video inputs. In Sec.3.1, we first introduce the data format of AlphaTablets, then discuss the differentiable rasterization of AlphaTablets in Sec.3.2. Finally, in Sec.3.3, we introduce a bottom-up pipeline based on AlphaTablets to conduct 3D planar reconstruction from monocular video input.

### 3.1 AlphaTablets: Representing 3D Planes with Semi-Transparent Rectangular Primitives

As illustrated in Figure 1, our proposed AlphaTablet is shaped as a rectangle with 3D geometry properties and 2D in-tablet properties. The 3D geometry includes the tablet center point $p$, normal vector $\mathbf{n}$, up vector $\mathbf{u}$ and right vector $\mathbf{r}$. The normal, up and right vectors are orthogonal. The 2D in-tablet information contains a texture map $c$, an alpha channel $\alpha$, and a pixel range $(ru, rv)$ that indicates the 2D resolution of the texture and transparency map. The alpha channel $\alpha$ ensures that arbitrary shapes can be modeled by our tablet formation. Since the ratio of unit distance in 3D space and 2D in-tablet canonical space varies across different tablets, distance ratios $\lambda_u, \lambda_v$ of two directions on the texture map is also maintained for every tablet to acquire the tablet size in 3D space.

## 3.2 Differentiable Rasterization

As a generic 3D planar surface representation, efficient projection from 3D to 2D images is highly demanding for AlphaTablets. Therefore, we introduce the differentiable rasterization of AlphaTablets. The data format of AlphaTablets is the 3D rectangle soup with alpha channels, allowing us to easily adapt and utilize existing mesh-based efficient differentiable rendering frameworks, such as NVDiffrast [24] to composite and render an arbitrary number, shape, and position of tablets in a fully differentiable manner.

**Pseudo Mesh Construction.** We can leverage differentiable mesh rendering frameworks like NVDiffrast [24] by converting the tablets into pseudo meshes before each rendering pass. Note that this conversion is just used for the adaption of NVDiffrast. Given a single tablet $t_i$, we can convert it to two mesh triangle faces through the following process:

$$v_i = \begin{aligned} & [p_i - ru_i/\lambda_{u,i} - rv_i/\lambda_{v,i}, \ p_i - ru_i/\lambda_{u,i} + rv_i/\lambda_{v,i}, \\ & p_i + ru_i/\lambda_{u,i} + rv_i/\lambda_{v,i}, \ p_i + ru_i/\lambda_{u,i} - rv_i/\lambda_{v,i}] \end{aligned} \tag{1}$$

$$f_i = [[0, 1, 2], [0, 2, 3]] \tag{2}$$

Where $v_i$ represents the 3D vertex coordinates, and $f_i$ denotes the face indices, consistent with the general mesh definition. The texture and alpha maps of all tablets are tiled onto a global texture map according to their respective resolutions $(ru_i, rv_i)$. The specific tile location serves as the texture coordinates for the four vertices of each tablet.

**Multi-layer Rasterization.** As transparency information is used in AlphaTablets, the rasterization process in our approach requires rasterizing multiple layers through depth peeling to extract multiple closest surfaces for each pixel. Given the model-view-projection (MVP) matrix $M_k$ of a specific view k, the rasterization result of the $l$-th closest surface for the image pixel (i, j) can be formed as:

$$\mathcal{R}_l(M_k, i, j) = (u_{i,j}, v_{i,j}, tri_{i,j}) \tag{3}$$

where $u_{i,j}$ and $v_{i,j}$ are the barycentric coordinates within a triangle, and $tri_{i,j}$ is the triangle index. The color and alpha values $c(i, j)$ and $\alpha(i, j)$ are then acquired from the texture map using the two coordinates and the triangle index.

**Anti-aliasing for AlphaTablets.** Previous anti-aliasing techniques in rasterization predominantly work on non-transparent primitives without considering learnable alpha values of each face, yet alpha channel is a crucial component for AlphaTablets. An intuitive approach to incorporate alpha channels would be to conduct anti-aliasing for both texture and alpha within each rasterization layer. However, this straight-forward method does not work well on tablets with alpha channels. A simple counterexample is two overlapping planes with constant alpha values of 0 and 1, respectively. The rasterization results in two rasterized layers for the intersection pixel of the two planes. And the anti-aliasing would result in both layers having an alpha value below 1, causing incorrect transparency and colors at the intersected boundaries in the final alpha blending process.

To address this issue, we propose an anti-aliasing method for the semi-transparent primitives. Given a pixel through which the boundary lines of two planes pass, with the original colors $c_1$ and $c_2$, and alpha values $a_1$ and $a_2$, respectively, and anti-aliasing weights $w$ and $1 - w$, the process to obtain the anti-aliased color $c_{aa}$ is as follows:

$$c_{aa} = \frac{\alpha_1 c_1 w + \alpha_2 c_2 (1 - w)}{\alpha_1 w + \alpha_2 (1 - w)} \tag{4}$$

And the alpha value is not anti-aliased. The intuitive idea is that the blending weights of anti-aliasing should not only be determined by overlapping areas, but also take alpha values into consideration. For each rasterization layer, anti-aliasing can be further expressed using the following formula:

$$c_{aa} = \frac{AA(\alpha c)}{AA(\alpha)}, \quad \alpha_{aa} = \alpha \tag{5}$$

Where $\alpha$ and $c$ are the alpha and color values before anti-aliasing, $c_{aa}$ and $\alpha_{aa}$ are the alpha and color values after anti-aliasing, and $AA$ is the anti-aliasing function. We show an example figure 9 in appendix to demonstrate the clear improvements after the anti-alias formula correction.

**Alpha Composition.** Once we have multiple rasterized layers, we can stack them in depth order and blend them using the alpha compositing process widely employed in volume rendering and neural

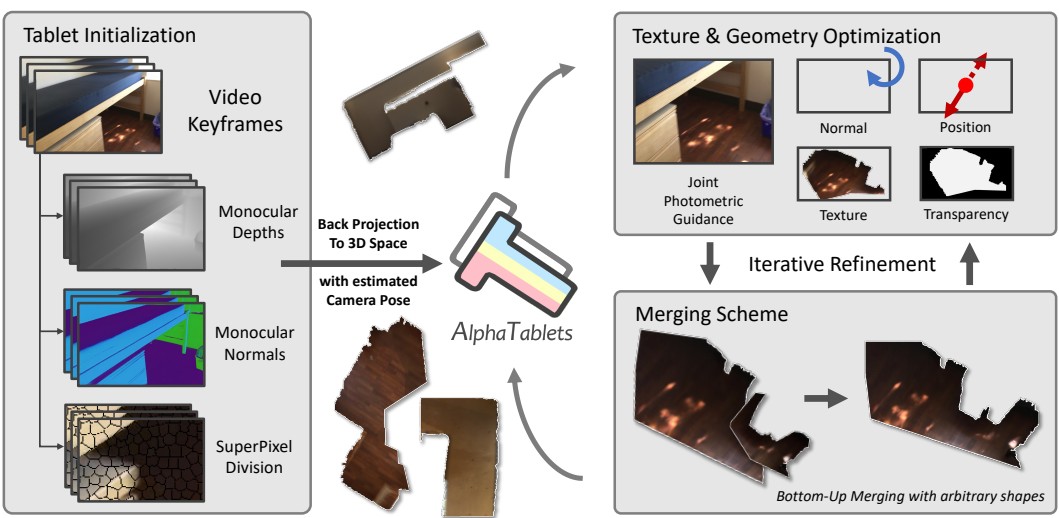

Figure 2: **Pipeline of our proposed 3D planar reconstruction.** Given a monocular video as input, we first initialize AlphaTablets using off-the-shelf superpixel, depth, and normal estimation models. The 3D AlphaTablets are then optimized through photometric guidance, followed by the merging scheme. This iterative process of optimization and merging refines the 3D AlphaTablets, resulting in accurate and complete 3D planar reconstruction.

radiance fields. The final color of pixel (i, j) can be calculated as:

$$c_{i,j} = \sum_{l=1}^{L} T_{i,j,l} \alpha_{i,j,l} c_{i,j,l}, \quad T_{i,j,l} = \prod_{m=1}^{l-1} (1 - \alpha_{i,j,m}) \tag{6}$$

Where $c_{i,j,l}$ and $\alpha_{i,j,l}$ are the color and alpha values at pixel $(i, j)$ of the $l$-th rasterization layer. $T_{i,j,l}$ represents the accumulated transmittance of the previous $l - 1$ layers for the given pixel.

### 3.3 A Bottom-up Planar Reconstruction Pipeline with AlphaTablets

AlphaTablets provide flexible 3D plane modeling and efficient differentiable rendering. Building on this, we propose a bottom-up 3D planar reconstruction pipeline for monocular video input. As illustrated in Figure 2, AlphaTablets are first initialized via off-the-shelf geometric estimations, then the texture and geometry parameters of AlphaTablets are optimized using differentiable rendering. Tablets are examined and merged towards larger and more complete 3D planes. The optimization and merging benefit from each other through iterative refinement. By formulating the plane segmentation as a bottom-up clustering of 3D AlphaTablets and plane parameter estimation as rendering based optimization, our pipeline can accurately reconstruct detailed planar surfaces.

**Initialization.** We initialize the AlphaTablets using off-the-shelf geometric prediction models, including those for depth, surface normal and superpixel. Specifically, for each keyframe of the input video, we first apply SLIC [1] method to segment images into 2D superpixels based on local color homogeneity. Next, we utilize pretrained geometric models [19, 12] to estimate depth and surface normal for each image. To obtain initial depth and surface normal values for each superpixel, we perform 2D average pooling within each superpixel's region. These 2D superpixels are then back-projected into 3D space to form the initial tablet representation. Besides 3D geometry, the texture maps and alpha channels are initialized using 2D pixel colors and superpixel masks, respectively. The rectangle bounding box is determined by the minimum and maximum values on each 3D tablet's two orthogonal axes: up and right vector.

**Optimization.** Initialized from 2D view-based depth, surface normal and superpixel estimations, the initial 3D AlphaTablets may contain errors, overlaps, and inconsistency. We thus perform differentiable rendering based optimization to update the parameters of 3D AlphaTablets.

*Learnable Parameters.* While the 3D AlphaTablets offer significant flexibility, directly optimizing them for unrestricted movement in 3D space can cause instability. To mitigate this, we constrain each tablet's center to remain on its initial camera ray. In this way, we thus optimize the normal vector **n**

and distance $d$, rather than 3D center position $p$, where $d$ represents the distance between the tablet's center and the camera center of the view from which it was initialized. The up vectors of tablets are updated with normal to keep the rigid transformation characteristics of the tablet. Additionally, the texture and alpha channel values of each tablet are treated as learnable parameters, enabling appearance refinement to enhance the fidelity of the reconstructed planar surfaces.

*Loss Design.* The optimization process is driven by a set of carefully designed loss functions that collectively refine the tablet parameters to achieve accurate planar reconstruction. Given input monocular video, we adopt the photometric loss as the mean squared error between the rendered image $c$ of the tablets and the observed input images $c_{gt}$: $\mathcal{L}_{pho} = ||c - c_{gt}||_2^2$. By minimizing the photometric loss, the 3D AlphaTablets can be optimized to better fit the input images. However, due to the ambiguity of photometric alignment and the complexity of optimization, updating AlphaTablets only with photometric loss results in fuzzy reconstructions. We thus introduce several important losses to help mitigate the ambiguity and regularize the reconstruction.

Specifically, we use alpha inverse loss to prevent the emergence of semi-transparent regions after alpha composition, which is defined as $\mathcal{L}_{ainv} = \prod_{l=1}^{L}(1 - \alpha_l)$. Moreover, we observed that multiple semi-transparency tablets, instead of one solid tablet, may occupy the same surface region to blend the rendering, which harms the geometric surface reconstruction. Inspired by mip-NeRF-360 [5], we utilize the distortion loss for AlphaTablets to penalize the multiple semi-transparency surfaces and promote the merging of tablets that are in close proximity. The distortion loss is defined as below:

$$\mathcal{L}_{dist} = \sum_{i=1}^{L-1} T_i T_{i+1} ||p_i - p_{i+1}||_2 \tag{7}$$

Where $T_i$ is the blending weight of the $i$-th rasterization layer defined in Sec. 3.2, $p_i$ is the 3D intersection point of the $i$-th rasterization layer. To further regularize the surface geometry and smoothness, we render the tablets to get the depths $d_r$ and surface normal maps $\mathbf{n}_r$, and supervise them by the input monocular depth and surface normal estimations $d_m, \mathbf{n}_m$ with mean squared error:

$$d_r = \sum_{l=1}^{L} T_l \alpha_l d_l, \quad \mathbf{n}_r = \sum_{l=1}^{L} T_l \alpha_l \mathbf{n}_l \tag{8}$$

$$d = \frac{d_r}{\prod_{l=1}^{L}(1 - \alpha_l)}, \quad \mathbf{n} = \frac{\mathbf{n}_r}{||\mathbf{n}_r||_2} \tag{9}$$

$$\mathcal{L}_{depth} = ||d - d_m||_2^2, \quad \mathcal{L}_{normal} = ||\mathbf{n} - \mathbf{n}_m||_2^2 \tag{10}$$

The final objective is defined as:

$$\mathcal{L} = w_1 \mathcal{L}_{pho} + w_2 \mathcal{L}_{ainv} + w_3 \mathcal{L}_{dist} + w_4 \mathcal{L}_{depth} + w_5 \mathcal{L}_{normal} \tag{11}$$

where $w_1, w_2, w_3, w_4, w_5$ are hyperparameters to balance the losses.

**Merging Scheme.** The optimized 3D AlphaTablets are still describing local 3D planar surface, bounded by the 2D superpixels. Therefore, to represent the exact 3D planes, we need to coherently merge the individual tablets into larger tablets. We introduce a hierarchical merging strategy that considers hybrid information including color, distance, and normal, to prevent the wrong merging.

Specifically, we first construct a KD-tree to find the tablet neighborhoods, and initialize a union-find data structure for all tablets. Each union-find set dynamically maintains the average color, center, and normal of all its constituent unit tablets. For each tablet, we search the KD-tree to find the K nearest tablets and evaluate whether they satisfy the following constraints for merging: (1) The angle between the normals of the two tablets should be smaller than a threshold $\theta$. (2) The angle between the average normals of the union-find sets to which the two tablets belong should be smaller than a threshold $\theta_s$. (3) The projected distance between the average centers of the union-find sets along their average normal should be smaller than a threshold $d$. (4) The difference between the average colors of the union-find sets should be smaller than a threshold c.

If all these constraints are satisfied, the two tablets are merged into the same union-find set. We repeat this process for all tablets, continually updating the average color, center, and normal of each union-find set as merges occur. This iterative merging procedure continues until no further merges are possible, resulting in a set of coherent planar surfaces represented by the final merged tablets. More details about merging are included in the appendix.

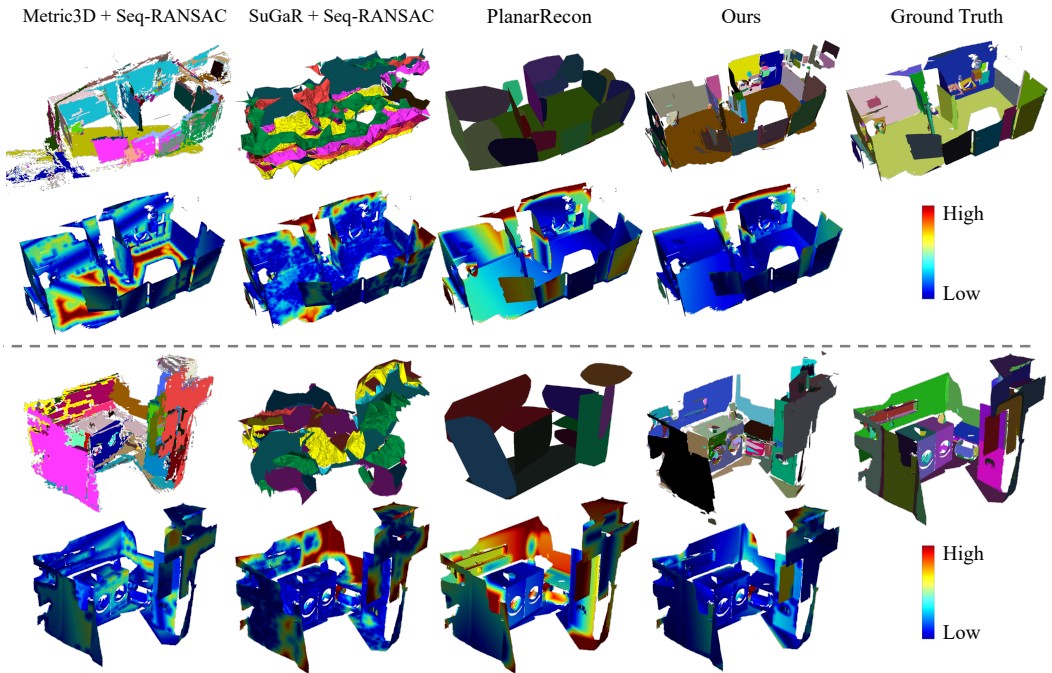

Figure 3: **Qualitative results on ScanNet.** Error maps are included. Better viewed when zoomed in.

Table 1: **3D geometry reconstruction results on ScanNet.**

| Method | Comp ↓ | Acc ↓ | Recall ↑ | Prec ↑ | F-Score ↑ |
|---|---|---|---|---|---|
| NeuralRecon [46] + Seq-RANSAC | 0.144 | 0.128 | 0.296 | 0.306 | 0.296 |
| Atlas [31] + Seq-RANSAC | 0.102 | 0.190 | 0.316 | 0.348 | 0.331 |
| ESTDepth [29] + PEAC [13] | 0.174 | 0.135 | 0.289 | 0.335 | 0.304 |
| PlanarRecon [50] | 0.154 | **0.105** | 0.355 | 0.398 | 0.372 |
| Metric3D [19] + Seq-RANSAC | **0.074** | 0.379 | 0.426 | 0.161 | 0.231 |
| SuGaR [18] + Seq-RANSAC | 0.121 | 0.324 | 0.385 | 0.296 | 0.327 |
| Ours | 0.108 | 0.161 | **0.481** | **0.447** | **0.456** |

## 4 Experiments

### 4.1 Evaluation on 3D Plane Detection and Reconstruction

**Implementation Details.** We use Metric3Dv2 [19] for predicting monocular depths and Omnidata [12] for surface normals. We leverage the keyframe selection method in NeuralRecon [46] to segment the scene into multiple parts. Each part undergoes separate optimization, followed by joint optimizations. The keyframe number of each part is set to 9. The separate optimization for each part is performed for 32 epochs, while the joint optimization step is executed for 9 epochs. The weights for the loss functions are set as follows: $[w_1, w_2, w_3, w_4, w_5] = [1.0, 1.0, 20.0, 4.0, 4.0]$. We use Adam optimizer, and the learning rates for the tablet's texture, alpha, normal, and distance are set to 0.01, 0.03, 1e-4, and 5e-4, respectively. After the second merge step, the learning rate for the distance is reduced to 2e-4. The normal threshold is set to 0.93. The entire reconstruction process for a single scene takes around 2 hours on average when executed on a single A100 GPU.

**Dataset and Evaluation Metrics.** We use ScanNetv2 [11] dataset to evaluate the 3D plane detection and reconstruction performance of our proposed method. Following PlanarRecon [50], our method is tested on the validation set of Atlas [31] using generated 3D plane ground truth. For evaluation metrics, we follow previous works [50, 26, 54] to use Murez's 3D metrics [31] for geometry, and rand index (RI), variation of information (VOI), segmentation covering (SC) as plane segmentation metrics. To assess the segmentation performance, the reconstructed plane instances are transferred onto the ground truth planes using the nearest neighborhood approach, following common practices.

Table 2: **3D plane segmentation results on ScanNet.**

| Method | VOI ↓ | RI ↑ | SC ↑ |
|---|---|---|---|
| NeuralRecon [46] + Seq-RANSAC | 8.087 | 0.828 | 0.066 |
| Atlas [31] + Seq-RANSAC | 8.485 | 0.838 | 0.057 |
| ESTDepth [29] + PEAC [13] | 4.470 | 0.877 | 0.163 |
| PlanarRecon [50] | 3.622 | 0.897 | 0.248 |
| Metric3D [19] + Seq-RANSAC | 4.648 | 0.862 | 0.209 |
| SuGaR [18] + Seq-RANSAC | 5.558 | 0.775 | 0.082 |
| Ours | **3.468** | **0.928** | **0.273** |

**TUM Dataset**        **Replica Dataset**

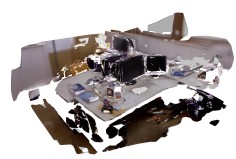 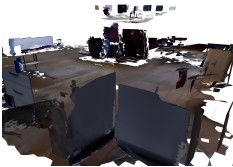 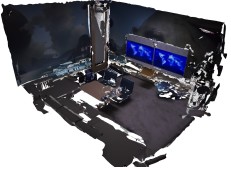 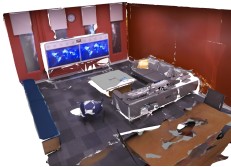

Figure 4: **Qualitative results on TUM-RGBD and Replica datasets.**

**Baselines.** We compare our method with different types of representative works. PlanarRecon [50] is the state-of-the-art method of learning-based 3D planar reconstruction from monocular video. Following it, we compare with strong baselines that first reconstruct the 3D scene and then fit 3D planes using RANSAC, including single or multi-view depth-based methods [29, 19], 3D volume-based methods [46, 31], and the recent 3D gaussian based method [18].

**Quantitative Results.** The evaluation results for 3D plane geometry and segmentation are presented in Table 1 and Table 2, respectively. Our proposed method achieves clear improvements compared to other state-of-the-art approaches across various evaluation metrics. 3D volume-based reconstruct-then-fit methods suffer from reconstruction errors and noise threshold-sensitive RANSAC, and exhibit relatively low performance in terms of planes' geometry and 3D coherence. Depth-based methods inherently encounter 3D inconsistency, resulting in fragmented and multi-layered predictions. PlanarRecon [50], which is specially trained on the ScanNet dataset, demonstrates the capability to predict major planes with high geometric accuracy. However, its performance is hindered by the limited coverage of the predicted planes and the failure to detect many small plane instances. Approaches based on 2D Gaussian Splatting [18] tend to be heavily influenced by the poor initialization, textureless regions and motion blurs in ScanNet, resulting in degraded reconstruction performance. Compared to other methods, our approach demonstrates much improved overall performance for both geometry and segmentation.

**Qualitative Results.** To provide a qualitative assessment of our method's performance, we follow PlanarRecon [50] and present the plane reconstruction results in Figure 3, along with the recall error maps. The SuGaR+Seq-RANSAC method suffers from erroneous geometric reconstructions, and the Metric3D+Seq-RANSAC is constrained by inconsistent fuzzy points and sub-optimal plane segmentations. PlanarRecon, while capable of reconstructing large planar surfaces with high geometric accuracy, struggles to capture and reconstruct smaller plane instances, resulting in incomplete representations of the scene. Our method benefits from the bottom-up planar reconstruction scheme, and can accurately predict the 3D plane instances while excelling in detecting and reconstructing details, particularly for smaller plane instances. This capability significantly outperforms other methods in handling fine-grained planar structures. To demonstrate the generalization ability of our method, we further test it on TUM-RGBD [45] and Replica [44] datasets. Qualitative results are shown in Figure 4. Our method can faithfully reconstruct 3D planar surfaces in various scenarios.

## 4.2 Ablation Studies

To validate the efficacy of our method's design, we conducted a series of ablation experiments exploring the impact of various components, including the loss functions, merge schemes, and tablet anti-aliasing. The results are presented in Table 3. Tablet distortion loss encourages the planar surfaces to converge and merge, leading to improved performance. Furthermore, the normal loss and depth loss significantly contribute to the geometric accuracy of the reconstructed planes, particularly

Table 3: **Ablation studies.** *AlphaInv* denotes the alpha inverse loss.

| Method | F-score ↑ | VOI ↓ | RI ↑ | SC ↑ |
|---|---|---|---|---|
| Only Photometric and *AlphaInv* loss | 0.240 | 4.096 | 0.936 | 0.191 |
| + Tablet Distortion loss | 0.271 | 3.741 | 0.937 | 0.253 |
| + Normal loss | 0.425 | 3.490 | **0.944** | 0.263 |
| + Depth loss | **0.456** | **3.466** | **0.944** | **0.284** |
| w/o tablet anti-aliasing | 0.415 | 3.545 | 0.937 | 0.280 |
| w/o tablet merge | 0.188 | 6.991 | 0.939 | 0.098 |
| Full | **0.456** | **3.466** | **0.944** | **0.284** |

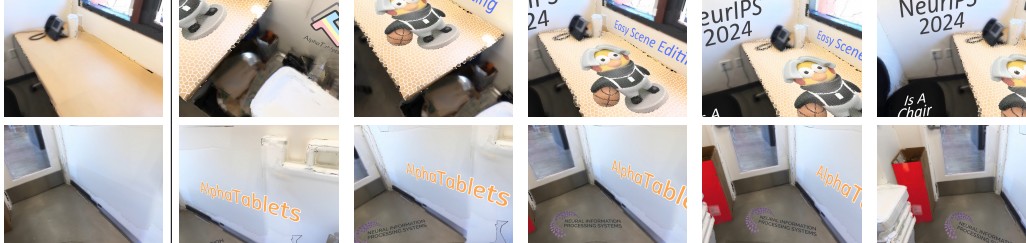

Original Scene            3D Coherent Scene Editings

Figure 5: **3D scene editing examples of our method.**

in textureless regions where photometric loss constraints are insufficient. Merging scheme is crucial for producing appropriate 3D planes. Without merging, the 3D AlphaTablets remain small plane fragments, and thus can not reconstruct 3D planes, as shown in Table 3. Moreover, tablet antialiasing contributes to smoother results, leading to enhanced overall performance.

### 4.3 Example Application: 3D Plane-based Scene Editing

One of the significant advantages of AlphaTablets representation is its ability to perform consistent 3D scene editing by simply modifying the 2D texture maps associated with the reconstructed planes. As illustrated in Fig. 5, our method can achieve impressive results for editing 3D scenes. The accurate plane reconstruction allows for precise texture mapping, enabling the seamless application of textures, text, or other visual elements onto the planar regions within the scene. Furthermore, our method offers the flexibility to modify the color or perform style transfer on individual planes, providing a powerful tool for creative scene manipulation.

### 4.4 Limitations and Future Work

AlphaTablets effectively represent 3D planes, but it may struggle in highly non-planar regions where the planar assumption for a single superpixel does not hold. Additionally, the current AlphaTablets representation does not account for view-dependent effects. As a result, the optimization relies on color consistency across views, which can be compromised by non-Lambertian surfaces or changes in lighting. In the future, we aim to enhance AlphaTablets with view-dependent modeling, and explore hybrid scene representation such as AlphaTablets with Gaussians.

## 5 Conclusion

In this work, we introduce AlphaTablets, a novel and versatile 3D plane representation. AlphaTablets use rectangles with alpha channels to represent 3D planes, allowing for flexible and effective arbitrary 3D plane modeling. We derive a differentiable rasterization process for AlphaTablets to enable efficient 3D-to-2D rendering. Building on this, we propose a novel bottom-up 3D planar reconstruction pipeline from monocular video input. Leveraging the AlphaTablets representation, along with carefully designed optimization and merging schemes, our pipeline can reconstruct highly accurate and complete 3D planar surfaces in a generalizable manner. Experiments on the ScanNet dataset demonstrate significant improvements over baseline methods, highlighting the potential of AlphaTablets as a general representation for 3D planar surfaces.

# Acknowledgement

This work was supported by Beijing Science and Technology plan project (Z231100005923029), the Natural Science Foundation of China (Project Number 62332019) and Beijing Natural Science Foundation (L222008).

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

# A Appendix

## A.1 More Details of AlphaTablets Optimization

**Update of up vector.** Note that the tablet's up vector should not be learned during the optimization process. By considering the tablet's motion as a rigid transformation, any in-plane rotation can be accounted for by optimizing the texture and alpha values. However, we need to establish an update rule for the up vector to keep the rigid transformation characteristics of the tablet. Our design is to apply the same rotation to the up vector as the one applied to the normal vector during the update. Given the normal vectors $n$ and $n'$ (before and after the update), and the up vector $u$, we can acquire the new up vector $u'$ by:

$$\theta_r = \mathbf{n} \cdot \mathbf{n}', \tag{12}$$

$$\mathbf{r} = \mathbf{n} \times \mathbf{n}', \tag{13}$$

$$K = \begin{pmatrix} 0 & -r_3 & r_2 \\ r_3 & 0 & -r_1 \\ -r_2 & r_1 & 0, \end{pmatrix} \tag{14}$$

$$R = I \cos \theta_r + K \sin \theta_r + \mathbf{rr}^T (1 - \cos \theta_r), \tag{15}$$

$$\mathbf{u}' = R\mathbf{u} \tag{16}$$

**Merging Scheme.** As explained in Section 3.3, we first construct a KD-tree to find the tablet neighborhoods, and initialize a union-find data structure for all tablets. Here we actually use the unit tablets, which are defined as the projections of all initial 3D tablets to the current tablets, to build the KD-tree. In other words, we maintain the affiliations of initial and current tablets, and use the updated initial tablets to perform the merging. The reason is that using 3D center point distances among tablets with different 3D sizes is ambiguous. For example, a small tablet can have a smaller center point distance than a non-adjacent small tablet, compared to a spatially adjacent big tablet. Using the unit tablets with similar sizes, the neighboring adjacency can be easily determined by checking the center point distances.

These unit tablets are used only because they have easily defined neighborhoods. Since the unit tablets of one current tablet come from the projection on this tablet, they share the same surface normal and adjacent positions. Thus, merging with unit tablets will definitely produce larger (or the same) tablets than current tablets. After each merging, all the unit tablets are updated by the projection onto the merged new tablets, and the merged new tablets are fed into the next optimization.

Using SLIC superpixel on ScanNet 1296x968 resolution image results in around 10k superpixels for each keyframe, leading to a large number of initial tablets. To address the issue, We conduct an initial merge process after AlphaTablets initialization. In practice, we find it is beneficial to the accuracy and convergence speed. Table 5 shows the ablations of the initial merge.

Table 4: **Ablation studies on initial merge.**

| Method | F-score ↑ | VOI ↓ | RI ↑ | SC ↑ |
|---|---|---|---|---|
| w/o in-training merge and init merge | 0.188 | 6.991 | 0.939 | 0.098 |
| w/o in-training merge | 0.438 | 5.171 | 0.941 | 0.138 |
| w/o init merge | 0.454 | 3.754 | **0.944** | 0.273 |
| w/ all merge schemes | **0.456** | **3.466** | **0.944** | **0.284** |

**Weight Check Scheme.** During the optimization process, there may be cases where some tablets are nearly invisible from all viewpoints, yet they have a relatively large transparency value. In such situations, these tablets should be removed. Additionally, there could be instances where certain regions of a tablet are not visible from any viewpoint. In these cases, those specific regions of the tablet should be excluded.

To address these scenarios, we designed a weight check mechanism: We perform a rasterization step at all viewpoints and extract the points where the alpha blending weight exceeds a certain threshold (we choose 0.3 in our implementation). We record the tablet index corresponding to each of these points. Before the merging step, we perform the weight check by removing tablets with an excessively

low number of associated points. Furthermore, for each tablet, we recalculate its boundary based on the texture map coordinate ranges of all the points associated with that tablet.

**Tablet-camera assignment.** We always maintain affiliations between the initial tablets and the current (merged) tablets (as stated in Sec A.1), and we keep track of the camera index that initially generated each initial tablet. When tablets are merged, we count the number of each camera index corresponding to all affiliated initial tablets and assign the most frequently occurring camera to the newly merged tablet.

## A.2    Additional Implementation Details

**Baselines.** For 3D volume-based methods including Atlas, NeuralRecon, PlanarRecon, and Metric3D with TSDF fusion, we followed PlanarRecon to use their enhanced version of Seq-RANSAC. We refer to PlanarRecon for detailed descriptions. For point-based methods such as SuGaR, since PlanarRecon's Seq-RANSAC requires 3D TSDF volume as inputs and cannot be easily adapted to points or meshes, we use the classical vanilla Seq-RANSAC, which iteratively applies RANSAC to fit planes. Here we used Open3D plane RANSAC implementation for each iteration. The hyper-parameters are carefully tuned for optimal performance. For the Metric3D baseline, We used the official Metric3D v2 implementation and pre-trained weights (v2-g) running on each keyframe to get depth maps, followed by TSDF fusion to fuse into 3D volume. Finally, PlanarRecon's Seq-RANSAC is applied to the 3D TSDF volume to get the planar results. We adopted the original implementation for the SuGaR baseline, during the COLMAP pre-processing, we feed ground-truth camera poses into the pipeline, which provides better initial sparse points. After optimization, SuGaR outputs the mesh model, and we uniformly sampled 100k surface points and applied vanilla Seq-RANSAC on top of sampled points to get the 3D planar results. Quantitative results for other baselines (Atlas, NeuralRecon, PEAC, PlanarRecon) were taken from PlanerRecon.

**Details of tablet properties.** For pixel range $(r_u, r_v)$, each tablet's geometry is located in 3D space, while its texture is stored in 2D. The pixel range represents the resolution at which the texture is stored: it is derived directly from the range in the source image for initial tablets; for merged tablets, the pixel range is calculated as the average of all corresponding initial tablets. The distance ratios $(\lambda_u, \lambda_v)$ establish the relationship between the 2D texture resolution and the 3D size of the tablet. For initial tablets, the distance ratio is calculated by dividing the camera's focal length by the average initial distance of the tablet. For merged tablets, the distance ratio is the average of all corresponding initial tablets' ratios. The alpha channel of tablets is a learnable single-channel map with the same shape as the texture map.

## A.3    Additional Discussions

**Different initialization for SuGaR baseline.** We further experiment on our ablation subset to compare the COLMAP initialization with Metric3D's dense depth-based initialization similar to our method. For Metric3D init, we use the same keyframes as our method and randomly sample a total of 100,000 points as initial points. The results are shown in the table:

Table 5: **Ablation studies on different initialization of SuGaR.**

| Method | F-score ↑ | VOI ↓ | RI ↑ | SC ↑ |
|---|---|---|---|---|
| SuGaR+COLMAP Initialization | 0.300 | 5.759 | 0.797 | 0.090 |
| SuGaR+Metric3D Initialization | 0.326 | 5.670 | 0.789 | 0.102 |
| Ours | **0.456** | **3.466** | **0.944** | **0.284** |

The ScanNet dataset presents significant challenges like numerous blurry and textureless regions, which are especially problematic for Gaussian-based methods like SuGaR when reconstructing clear geometry. Also, SuGaR heavily relies on COLMAP reconstruction to initialize, but the COLMAP reconstruction on ScanNet is sometimes noisy, affecting the final performance. The Metric3D initialization method does indeed enhance the reconstruction quality of SuGaR (as shown in Fig. 6), but the overall reconstruction quality remains constrained, with noticeable jitter and challenges in accurately delineating planar regions, leading to an inferior performance to our approach.

SuGaR with COLMAP Initialization     SuGaR with Metric3D Initialization

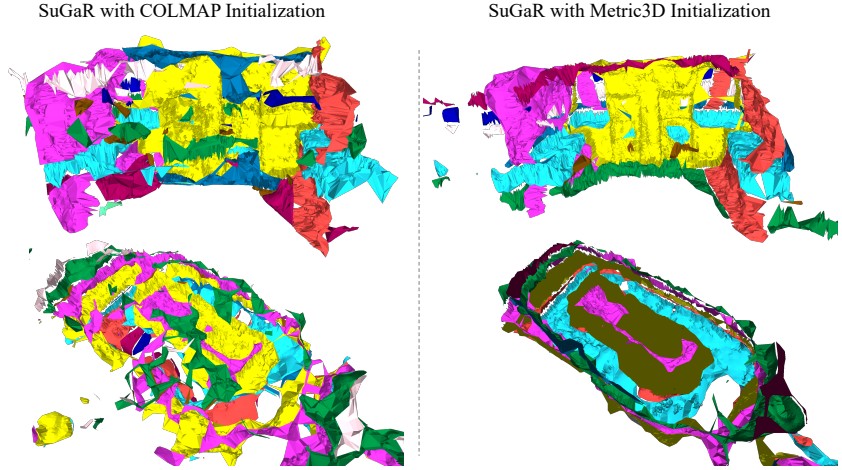

Figure 6: **Qualitative Comparison of Initialization Methods for SuGaR.**

**Breakdown of Time Budget.** Below is a breakdown of the time budget for the optimization process of a single scene:

Table 6: **Breakdown of the time budget of a single scene.**

| Stage | Task | Time (s) |
|---|---|---|
| **Initialization** | texture init | **1517.38** |
| | geometry init | **1672.57** |
| **Render** | pseudo mesh | 10.39 |
| | rasterization | 316.62 |
| | alpha composition | 2.15 |
| **Loss Calculation** | photometric loss | 1.07 |
| | depth loss | 28.28 |
| | normal loss | 102.44 |
| | distortion loss | 5.90 |
| **Training** | backward | **3347.83** |
| **Merge** | kd-tree,union-find set | 96.41 |
| | geometric calculation | 23.14 |
| | tablet projection | 22.26 |
| | weight check | 62.14 |

The merge and rendering pipeline is relatively efficient, while the initialization process (which includes converting every superpixel to an initial tablet, and texture initialization) consumes a significant amount of time. This is due to the current naive demonstration implementation, where tens of thousands of Python loops are called, which can be improved to enable parallelized initialization in future work. Furthermore, the NVDiffRast renders more than ten layers to perform alpha composition every forward pass, but most of the scene's structure is single-layered, resulting in a substantial backward computation burden during training. We regard this as another potential area for considerable optimization in the future work.

**3D reconstruction accuracy.** The difference in 3D accuracy (termed as Acc in Tab. 1 of the main paper) between our method and PlanarRecon on the ScanNet dataset can be attributed to several factors. First is the scope of reconstruction: PlanarRecon often only reconstructs large planar regions. This allows for easier localization and high accuracy in these specific areas, but it limits overall coverage and performance. Our method enables more comprehensive reconstruction, including smaller planar regions, which can impact the accuracy metrics but provide a more complete representation of the scene. Another is the ground-truth coverage: It is worth noting that the 3D ground truth planes in ScanNet only partially cover the scene within the camera's view. Even after

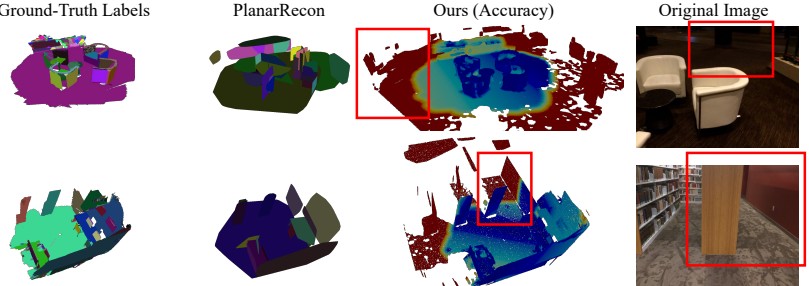

| Ground-Truth Labels | PlanarRecon | Ours (Accuracy) | Original Image |

*Red area denotes low accuracy area, including many regions **indeed exist** but did not appear in the ground-truth.
*Red rectangles show regions reconstructed by our method but did not included by ground-truth.

Figure 7: **Demonstration of Insufficient Coverage of 3D Ground-Truth Labels**: The 3D ground truth labels only partially cover the range within the camera's view. Most of the red regions in the figure highlight this issue. While these uncovered areas reduce accuracy, they should not be considered a negative outcome.

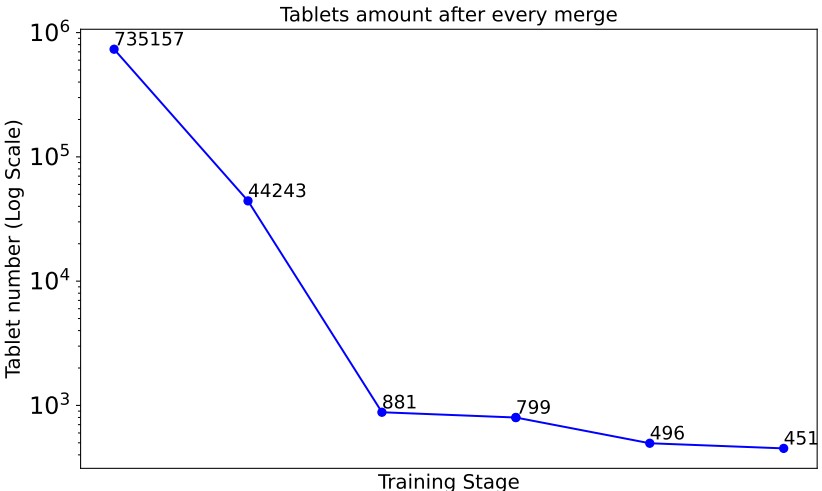

Figure 8: **Visualization of Tablet Count Evolution.**

excluding areas too distant to be relevant using the camera frustum, significant portions remain uncovered. PlanarRecon learns to exclude distant reconstructions during its training stage, leading to improved accuracy metrics. Our method, however, is capable of identifying planar regions for all visible areas (evident in Fig. 7 where most of the red regions highlight this phenomenon). While these uncovered areas affect the evaluation accuracy, they should not necessarily be considered a negative outcome. Our method provides a more complete reconstruction of the scene, including areas not represented in the ground truth data.

**Tablet count evolution.** We demonstrate the tablet count evolution of a single scene in Fig. 8. The number of tablets decreases rapidly in the early merging stages and gradually converges into several hundred. Notably, the final tablets contain a large portion of small tablets representing non-planar regions, while the primary planar scene structure is adequately represented with fewer tablets.

## A.4 Additional Qualitative Results

We provide more qualitative results in Fig. 10 and Fig. 11.

## A.5 Broader Impacts

3D planar reconstruction and editing have the potential to revolutionize numerous fields such as entertainment, media, accessibility, manufacturing, etc, by enhancing visualization, interaction, and

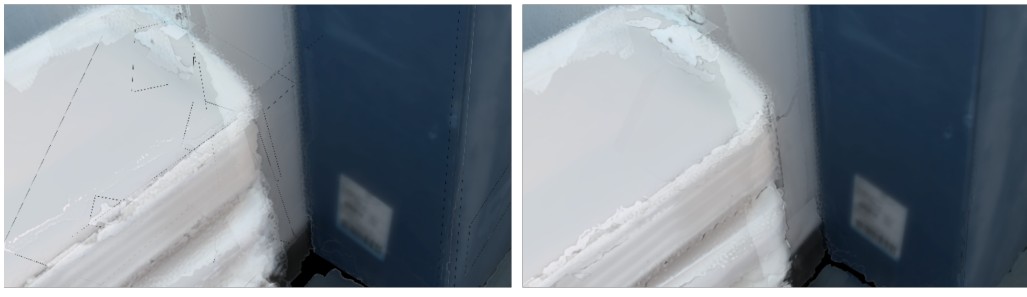

| Naive Anti-aliasing | Our Tablet Anti-aliasing |

Figure 9: **Qualitative comparison of our tablet anti-aliasing scheme.** Naive anti-aliasing will lead to wrong strip artifacts, while our anti-aliasing scheme effectively mitigates those artifacts.

understanding. However, it may raise concerns about privacy and data security, necessitating robust policies and safeguards.

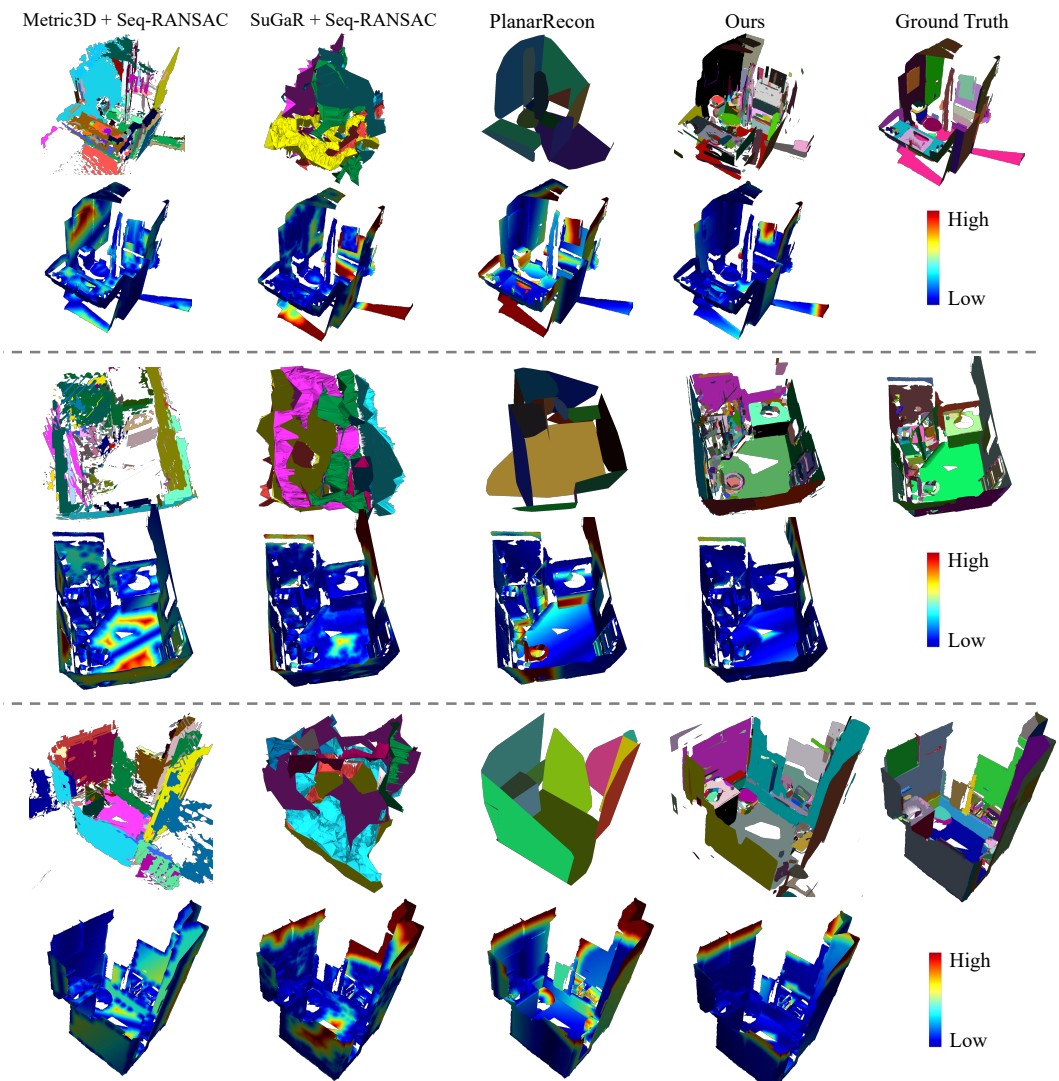

Figure 10: **More qualitative results on ScanNet.** Error maps are included. Better viewed when zoomed in.

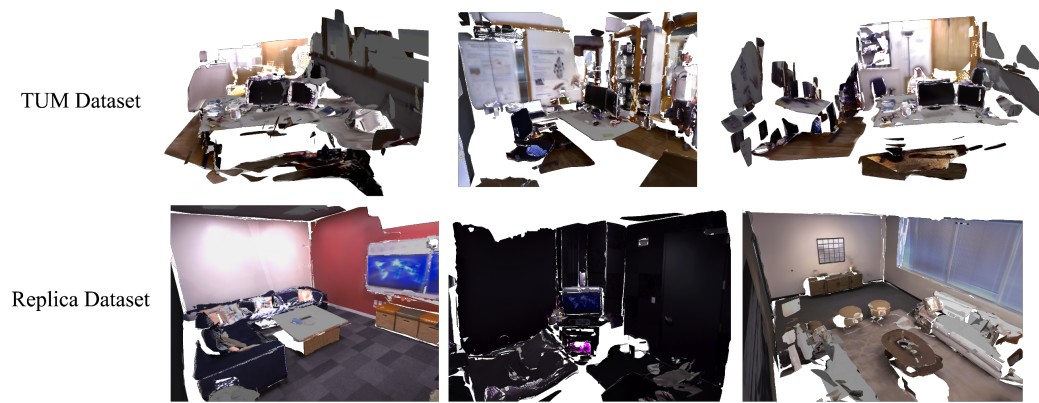

Figure 11: **More qualitative results on TUM-RGBD and Replica datasets.**

