# OpenReview forum: "AlphaTablets: A Generic Plane Representation for 3D Planar Reconstruction from Monocular Videos"
_NeurIPS.cc/2024/Conference — NeurIPS 2024 poster_

### Official Review · Reviewer_Ku7J · 2024-07-12

**Soundness:** 3
**Presentation:** 2
**Contribution:** 3
**Rating:** 6
**Confidence:** 5

**Summary:**

This paper introduces a novel and generic representation of 3D planes called AlphaTablets. AlphaTablets represent 3D planes as rectangles with alpha and RGB channels, enabling accurate, flexible, and consistent modeling. The paper also proposes a differentiable rasterization method on top of AlphaTablets, as well as a bottom-up planar reconstruction pipeline for the differentiable optimization of target 3D planes. Additionally, an effective merging scheme is introduced during optimization to facilitate the growth and refinement of AlphaTablets. The extensive experiments conducted on the ScanNet dataset demonstrate the state-of-the-art performance of AlphaTablets in 3D planar reconstruction.

**Strengths:**

1.	The concept of optimizing 3D planar primitives, referred to as AlphaTablets in this paper, through differentiable rendering is interesting. While previous methods such as PlanarRecon adopt a feedforward approach, training a model with 3D plane labels, this process can be costly and result in out-of-distribution issues when applied to unseen datasets. In contrast, reconstructing planar scenes using a differentiable rendering approach and solely relying on 2D supervision offers greater flexibility and potential.
2.	The planar reconstruction results achieved on the ScanNet dataset demonstrate state-of-the-art performance in both 3D geometry and 3D plane segmentation metrics, underscoring the excellence of this paper and the potential of the proposed pipeline.
3.	The coherent scene editing application is good.

**Weaknesses:**

1.	I have concerns regarding the speed of the proposed method. As stated in section 4.1, it takes approximately 2 hours to reconstruct a single scene. Could the authors provide details on the time taken for each part of the entire system during optimization? Additionally, where is the primary bottleneck for speed enhancement? A detailed efficiency analysis from the author would be greatly appreciated.
2.	Some details regarding the plane parameters are unclear to me. What are the values of the pixel range (ru, rv) in the paper, and are they the same for all tablets? How are the distance ratios $\lambda_u$ and $\lambda_v$ calculated? The alpha channel is described as a learnable single-channel map with the same shape as the texture map. Is my understanding correct?
3.	The initialization process is also unclear to me. It appears that each superpixel is initialized as a tablet at the beginning of the optimization. This could potentially introduce a large number of initial tablets across all 9 keyframes in a video segment, which may adversely impact the optimization speed. Why not merge similar superpixels to create larger initial planes?
4.	The 3D accuracy of this method on the ScanNet dataset is notably higher than that of PlanarRecon. What are the main reasons for the lower reconstruction accuracy of the proposed method?
5.	Which model did the author use in practice to predict the monocular depth and normals? Was it Metric3d or Omnidata? Additionally, the scale of the used monocular depth may not align well with the ground-truth depth. Employing mean squared error in eq (10) for depth loss could potentially introduce additional errors.

**Questions:**

See weakness.

**Limitations:**

The authors have discussed the limitations.

---

> ### Author Rebuttal · Authors · 2024-08-05
>
> Thanks for the comments. All responses below will be put into revision.
>
> **[W1: break down of time budget and analysis]**
>
> Below is a breakdown of the time budget for the optimization process of a single scene:
>
> | **Stage**            | **Task**               | **Time (s)** |
> | -------------------- | ---------------------- | ------------ |
> | **Initialization**   | texture init           | **1517.38** |
> |                      | geometry init            | **1672.57** |
> | **Render**           | pseudo mesh            | 10.39 |
> |                      | rasterization          | 316.62 |
> |                      | alpha composition      | 2.15 |
> | **Loss Calculation** | photometric loss       | 1.07 |
> |                      | depth loss             | 28.28 |
> |                      | normal loss            | 102.44 |
> |                      | distortion loss        | 5.90 |
> | **Training**         | backward               | **3347.83** |
> | **Merge**            | kd-tree,union-find set | 96.41 |
> |                      | geometric calculation  | 23.14 |
> |                      | tablet projection      | 22.26 |
> |                      | weight check           | 62.14 |
>
> As shown above, the merge and rendering pipeline is relatively efficient. The initialization process (which includes converting every superpixel to a initial tablet, and texture initialization) consumes a significant amount of time. This is due to the current naïve implementation for demonstration, where tens of thousands of Python loop is called. We will improve the implementation to enable parallelized initialization in the future work. Furthermore, the NVDiffRast renders more than ten layers to perform alpha composition every forward pass, but most of the scene's structure is single-layered, results in a substantial backward computation burden during training. We regard this as another potential area for considerable optimization in the future work.
>
> **[W2: detail of tablet parameters]**
>
> Thanks and we appreciate the opportunity to clarify these details:
>
> 1. Pixel Range (ru, rv):
>
> - Each tablet's geometry is located in 3D space, while its texture is stored in 2D.
> - The pixel range represents the resolution at which the texture is stored.
> - For initial tablets, the pixel range is derived directly from the range in the source image.
> - For merged tablets, the pixel range is calculated as the average of all corresponding initial tablets.
>
> 2. Distance Ratios (λu and λv):
>
> - These ratios establish the relationship between the 2D texture resolution and the 3D size of the tablet.
> - For initial tablets, the distance ratio is calculated by dividing the camera's focal length by the average initial distance of the tablet.
> - For merged tablets, the distance ratio is the average of all corresponding initial tablets' ratios.
>
> 3. Alpha Channel: Yes, the alpha channel is a learnable single-channel map with the same shape as the texture map.
>
> **[W3: Initial tablet merge]**
>
> During initialization, each superpixel is initialized as a tablet using the estimated depth and surface normal. Using SLIC superpixel on ScanNet 1296x968 resolution image results in around 10k superpixels for each keyframe, leading to a large number of initial tablets. As detailed in Appendix Sec A.1, we introduce an init merge to create larger initial tablets, which helps to improve the speed. Further efficiency improvements could be achieved through fewer superpixel initial number (by controlling the SLIC hyper-parameters) and more aggressive init merge.
>
> **[W4: reconstruction accuracy]**
>
> The difference in 3D accuracy (termed as Acc in Table 1 of the main paper) between our method and PlanarRecon on the ScanNet dataset can be attributed to several factors:
>
> 1. Scope of Reconstruction:
>
> - PlanarRecon often only reconstructs large planar regions. This allows for easier localization and high accuracy on these specific areas, but it limits overall coverage and performance.
> - Our method enables more comprehensive reconstruction, including smaller planar regions, which can impact the accuracy metrics but provides a more complete representation of the scene.
>
> 2. Ground Truth Coverage:
>
> - It is worth noting that the 3D ground truth planes in ScanNet only partially cover the scene within the camera's view. Even after excluding areas too distant to be relevant using the camera frustum, significant portions remain uncovered.
>
>   - PlanarRecon learns to exclude distant reconstructions during its training stage, leading to improved accuracy metrics.
>   - Our method, however, is capable of identifying planar regions for all visible areas. This is evident in Figure 2 of the attached PDF, where most of the red regions highlight this phenomenon.
>
>   - While these uncovered areas affect the evaluation accuracy, they should not necessarily be considered a negative outcome. Our method provides a more complete reconstruction of the scene, including areas that are not represented in the ground truth data.
>
> Please refer to Figure 2 of the attached PDF for a qualitative illustration of this issue.
>
> **[W5: depth loss and model]**
>
> To clarify, we use Metric3D v2 for predicting monocular depths and Omnidata for surface normals.
>
> Yes, there is a trade-off when using depth loss due to the depth estimation errors. To address this issue, we incorporate multiple loss terms, including photometric, normal and distortion losses, etc., to optimize and regularize the tablets. The depth loss could be viewed as a data term in optimization which constraints the results not too far from the initializations. In practice, we observed that the depth loss helps with the final reconstruction, and it is a common practice in 3D reconstruction to employ it. Further incorporation of uncertainty or better depth estimation methods could be beneficial, which we will put into future work.
>
> Once again, we thank the reviewer for the valuable comments that helps improve our paper.

---

> > ### Comment · Reviewer_Ku7J · 2024-08-13
> >
> > Good response! Thanks for the efforts of the authors. My concerns have been well addressed. I will improve my rating in the final decision.

---

> ### Comment · Reviewer_Ku7J · 2024-08-14
>
> Thanks for the efforts of the authors again. I have some more questions about the paper in the following:
> 1. How many scenes are used to evaluate on the ScanNetv2 datasets in practice?
> 2. Will the authors release the code?
> 3. I noticed a recent work for planar reconstruction called AirPlanes in CVPR2024. It seems that a good dense reconstruction + RANSAC is still a strong baseline for planar reconstruction. Can the author further discuss the advantages of plane-based/primitive-based optimization for this task? The authors only need to discuss and do not need to compare with Airplanes.

---

> > ### Author Response · Authors · 2024-08-14
> > **Thanks for your helpful review**
> >
> > Many thanks for your valuable reviews. Here are our responses.
> >
> >
> >
> > **[1. Number of scenes to evaluate]**
> >
> > We follow the evaluation settings used by PlanarRecon and Atlas. Our evaluation is conducted on the official validation set of ScanNet v2, which consists of 312 scenes.
> >
> >
> >
> > **[2. Code release]**
> >
> > Yes, our code will be released once the paper is public.
> >
> >
> >
> > **[3. Discussions of primitive-based optimization's advantages and Airplanes]**
> >
> > We appreciate the reviewer mentioning AirPlanes, a concurrent great work (not publicly visible when this paper is submitted) presented at CVPR 2024. While both approaches aim for 3D planar reconstruction task, our method offers several distinct advantages:
> >
> > 1. Explicit geometric structure preservation: Compared to point/voxel-based dense reconstruction, using plane/primitive as the basic units explicitly enforces the local geometric (planar) structure during the reconstruction process. Through optimization and merging, these explicit geometric constraints are extended to the entire planar structure. This geometric structure preservation is not trivial or even challenging for depth map / dense points representation, For example, the dense 3D reconstruction of a wall often contains local non-planar / non-smooth regions and noisy outlier points, which affect the subsequent plane label fitting and the resulting geometric accuracy, since RANSAC only responses for extracting the highest-scoring planes with no further optimizations. To the contrary, our planar primitive (AlphaTablets) based optimization pipeline is more straightforward and arguably more effective for the planar reconstruction task, with as-planar-as-possible guaranteed and optimization in the loop.
> > 2. Generic plane representation: We proposed AlphaTablets as a general 3D plane representation, with a 3D planar reconstruction pipeline demonstrating its effectiveness. In contrast, AirPlanes mainly focus on the modular design of 3D planar reconstruction system, where planes are represented with point groups. Our explicit continuous 3D plane representation enables advanced applications such as texture editing and novel view synthesis. Furthermore, if needed, the resulting 3D planar reconstruction of AirPlanes could be represented as AlphaTablets to enable further optimization w.r.t input images for both plane geometry and textures, which is not easy for point or mesh representations (optimization could break the planar structure).
> > 3. Better generalization without dataset-specific training: Our method can operate without the need for training on specific datasets, leading to better generalization across various datasets. In contrast, AirPlanes needs to be trained (especially the plane embedding) with datasets, which may limit its applicability to diverse scenarios without further fine-tuning.
> > 4. Complementary contributions: The main contribution of AirPlanes, embedding grouping, is orthogonal to our approach. Planar embeddings could serve as a similarity measurement and be integrated into our merging stage, suggesting exciting possibilities for future research.

---

> > > ### Comment · Reviewer_Ku7J · 2024-08-14
> > >
> > > Thanks for the quick response. I finally decided to improve my rating to WA.

---

### Official Review · Reviewer_ELQv · 2024-07-12

**Soundness:** 2
**Presentation:** 3
**Contribution:** 3
**Rating:** 5
**Confidence:** 3

**Summary:**

The paper presents a light-weight 3D scene representation, which utilizes oriented 2D rectangles in 3D space with associated 2D texture and alpha maps (AlphaTablets).

For the task of 3D indoor scene reconstruction and 3D plane decomposition from multi-view posed RGB images (keyframes of a monocular video), the method uses off-the-shelf depth estimation, normal prediction and super-pixel segmentation on each image to initialize a set of the proposed AlphaTablets in 3D space: Each predicted superpixel segment is backprojected to 3D using the predicted, averaged depth of the segment and the known camera information. The orientation, size and texture of the rectangle are initialized based to the predicted, averaged normals, the bounding box of the superpixel and its color, respectively.

These AlphaTablets are then optimized using differentiable rendering and iteratively merged into larger entities.
The differentiable rendering considers the transparency, the merging strategy employs several constraints.

Following baseline evaluation protocols, the paper outperforms current baseline methods on 3D plane segmentation metrics and several evaluated 3D geometry reconstruction metrics on the ScanNet validation set.

The paper shows that the proposed method allows for texture-based scene edits.

**Strengths:**

The paper presents a light-weight 3D scene representation combining oriented, textured 2D rectangles with an additional alpha mask. The representation and method (initialization and optimization) are described clearly. Figure 2 clearly depicts (most of) the inputs to the system and gives a good overview of the method. The optimization terms seem to be well thought out and ablated.
The approach also quantitatively outperforms baselines methods on several 3D metrics.

**Weaknesses:**

- The descriptions of the newly evaluated baseline methods (Metric3D, SuGaR) (L267-271) is very brief and incomplete, which makes it difficult to understand how they were adapted to the specific scenario, including a description of the Seq-RANSAC setup. Additionally, a reference in the text should be added to indicate that quantitative results of the other baselines were taken from PlaneRecon [49].

- Geometric evaluation: Following the evaluation of Atlas [30], to assess the geometry reconstruction quality, 3D points are sampled both on the ground truth mesh and the prediction. However, given that the proposed method employs alpha masks on the rectangles, it is unclear how points are sampled, i.e., does the point sampling considers the alpha mask? If not, this would lead to an inconsistency between the visible geometry and the evaluated geometry (points are sampled on the rectangles).

- While I appreciate the fact that authors included a video to visualize some of the results, the results look quite blurry in several areas of the reconstructions (the telephone in the desk, the borders of geometry). Could the authors discuss these results in more detail?

Minor:
- In Fig. 2 it would be helpful to indicate that for the backprojection also the keyframe's camera information is being used.
- Worth adding related work: Huang et al. 3DLite: Towards Commodity 3D Scanning for Content Creation (TOG, 2017)

**Questions:**

- The qualitative results of SuGaR look surprisingly bad, can the authors please explain the setup of this baseline in more detail, especially the number of views used? Is it possible to use the proposed AlphaTablet initialization method (backprojected superpixels segments) for the 2D Gaussians in SuGaR?

- Given that the optimization takes 2 hours per scene, (compared to 5sec and 60sec of PlanarRecon and NeuralRecon) it would be interesting to see a break down of the time budget, whether it is spend on the differentiable rendering, merging stage (KNN lookup), or due the evaluation of a large number of constraints on potentially many samples.

- After two tablets are merged, what is the assigned camera ray the newly merged tabled is allowed to optimize its distance to the camera ray?

- A plot of the number of tablets over the course of the optimization/merging would be worth including to get a sense of the magnitude of optimized elements.

**Limitations:**

The paper addresses some of its limitations. However, a discussion about the method's runtime and suggestions on how to improve it would be interesting.

---

> ### Author Rebuttal · Authors · 2024-08-05
>
> Thanks for the comments. All responses below will be put into revision.
>
> **[W1:Baseline setup]**
>
> Thanks and we appreciate the opportunity to clarify:
>
> 1. Seq-RANSAC: For 3D volume-based methods including Atlas, NeuralRecon, PlanarRecon, and Metric3D with TSDF fusion, we followed PlanarRecon to use their enhanced version of Seq-RANSAC. We refer to PlanarRecon for detailed descriptions.
>    For point-based methods such as SuGaR, since PlanarRecon’s Seq-RANSAC requires 3D TSDF volume as inputs and cannot be easily adapted to points or meshes, we use the classical vanilla Seq-RANSAC, which iteratively applies RANSAC to fit planes. Here we used Open3D plane RANSAC implementation https://www.open3d.org/docs/latest/tutorial/Basic/pointcloud.html#Plane-segmentation for each iteration. The hyper-parameters are carefully tuned for optimal performance.
>
> 2. Metric3D: We used the official Metric3D v2 implementation and pre-trained weights (v2-g) from https://github.com/YvanYin/Metric3D. The Metric3D is run on each keyframe to get depth maps, followed by TSDF fusion (widely adopted implementation from https://github.com/andyzeng/tsdf-fusion-python) to fuse into 3D volume. Finally, PlanarRecon’s Seq-RANSAC is applied to the 3D TSDF volume to get the planar results.
>
> 3. SuGaR: We adopted the original implementation from https://github.com/Anttwo/SuGaR. During the COLMAP pre-processing, we feed ground-truth camera poses into the pipeline, which provides better initial sparse points. After optimization, SuGaR outputs the mesh model, and we uniformly sampled 100k surface points and applied vanilla Seq-RANSAC on top of sampled points to get the 3D planar results.
>
> We will include these details and add references in the revision to indicate that quantitative results for other baselines were taken from PlanerRecon.
>
> **[W2:Geometric evaluation with alpha mask]**
>
> To clarify, our point sampling process considers the alpha mask during evaluation. Specifically, we first perform a weight check (as described in Appendix Sec.A.1) using alpha channels to mask out transparent or unseen regions from all views. Then we follow the PlanarRecon to convert the remaining visible plane regions into mesh using Delaunay triangulation. Points are then evenly sampled on this generated mesh. We refer to PlanarRecon for more details.
>
> **[W3:blurry results]**
>
> Since our method focuses on reconstructing 3D planar surfaces, it has limitations when dealing with geometrically complex, non-planar regions, as discussed in the limitation section. Also, our current AlphaTablets do not consider view-dependent effects, leading to visual quality affected. As a potential solution, the introduction of a hybrid representation that combines our planar AlphaTablets with other methods, such as 3D Gaussians, may be promising for representing complex, non-planar geometries, which we will discuss in future work in the revision.
>
> **[W4,5:figure&related work]**
>
> Thanks and we will revise the text as suggested.
>
> **[Q1:Details and init of SuGaR baseline]**
>
> We appreciate the opportunity to provide more details on our setup and results:
>
> - Baseline Setup: We utilized all available views when training SuGaR. We adopted the official implementation, and enhanced COLMAP sparse reconstruction with ground-truth camera poses, which provides better initial sparse points.
> - Performance discussion: The ScanNet dataset presents significant challenges like numerous blurry and textureless regions, which are especially problematic for Gaussian-based methods like SuGaR when reconstructing clear geometry. Also, SuGaR heavily relies on COLMAP reconstruction to initialize, but the COLMAP reconstruction on ScanNet is sometimes noisy, affecting the final performance.
> - Initialization Experiment: We experiment on our ablation subset to compare the COLMAP initialization with Metric3D’s dense depth-based initialization similar to our method. For Metric3D init, we use the same keyframes as our method and randomly sample a total of 100,000 points as initial points. The results are shown in the table:
>
> | Method              | FScore    | VOI       | RI        | SC     |
> | ------------------- | --------- | --------- | --------- | --------- |
> | SuGaR+COLMAP Init   | 0.300     | 5.759     | 0.797     | 0.090     |
> | SuGaR+Metric3D Init | 0.326     | 5.670     | 0.789     | 0.102     |
> | Ours                | **0.456** | **3.466** | **0.944** | **0.284** |
>
> The Metric3D init method does indeed enhance the reconstruction quality of SuGaR (qualitative results in Fig.1 in attached PDF file), but the overall reconstruction quality remains constrained, with noticeable jitter and challenges in accurately delineating planar regions, leading to an inferior performance to our approach.
>
> **[Q2:breakdown of time budget]**
>
> Please refer to response to **W1 of reviewer Ku7J**.
>
> **[Q3:Tablet-camera assignment]**
>
> To clarify, (1) we maintain affiliations between the initial tablets and the current (merged) tablets (as stated in Sec A.1); (2) We keep track of the camera index that initially generated each initial tablet; (3) When tablets are merged, we count the number of each camera indices corresponding to all affiliated initial tablets, and assign the most frequently occurring camera to the newly merged tablet. We will include these details in the revision.
>
> **[Q4:tablet number plot]**
>
> Thank you for the great suggestion. We plot the numbers in Fig.3 of the attached PDF file. The number of tablets decreases rapidly in the early merging stages and gradually converges into several hundreds. Notably, the final tablets contain a large portion of small tablets representing non-planar regions, while the primary planar scene structure is adequately represented with fewer tablets.
>
> **[Limitations]**
>
> Thanks and we will revise the paper with a detailed runtime analysis and potential improvements as discussed above.
>
> Once again, we thank the reviewer for the valuable comments that helps improve our paper.

---

### Official Review · Reviewer_VGFx · 2024-07-13

**Soundness:** 3
**Presentation:** 3
**Contribution:** 3
**Rating:** 7
**Confidence:** 5

**Summary:**

The paper presents a novel scene representation, AlphaTablets, for planar scene reconstruction. AlphaTablets are bounded plains with a texture map and an alpha channel, which can be optimized through a differentiable rendering scheme. By applying conventional photometric losses with regularization, AlphaTablets can be directly fit to posed monocular videos. Experiments show that the proposed method outperforms existing planar reconstruction baselines. Further, due to the texture map representation of AlphaTablets, they can be used to edit planar regions in 3D scenes.

**Strengths:**

1. Representing scenes with planar structures has many important applications in AR/VR. Despite the importance however, there aren't many papers directly tackling the problem (compared to papers solving related problems such novel view synthesis). The paper proposes a simple yet effective representation to perform 3D planar reconstruction, which I believe can foster further interest in the community for future research.

2. Quantitative evaluations are conducted with a proper set of baselines, and the results suggest that the proposed pipeline effectively outperforms the tested baselines.

3. The downstream application on 3D scene editing is interesting, and seems to have potential adaptations for AR/VR services.

4. The writing is clear to follow. The training scheme presented in Section 3.3 is straightforward and intuitive, quite akin to training procedures used for NeRFs or Gaussian splats. This is a rather good aspect in my opinion: because it indicates that the proposed AlphaTablets representation is effective on its own and does not require any significant modifications in the training setup for successful 3D reconstruction.

**Weaknesses:**

Overall I find the paper to propose a solid solution to 3D planar reconstruction. I only have a few minor suggestions.

1. Currently, the notion of 'rasterization layers' is not clearly stated in Section 3.2. I suggest improving Figure 1 to show how multiple layers of AlphaTablets are composed to render each pixels in an image.

2. The captions of figures lack information. The paper would be easier to follow if each figure was accompanied with a more detailed caption. For example, Figure 1 could elaborate more on what 'up vectors' are, or what 'distance ratios' in the figure means.

3. What is the size of the resulting AlphaTablets representation? One of the benefits of 3D planar representations is the low storage cost. I wonder if AlphaTablets are sufficiently light-weight in terms of storage.

**Questions:**

Please refer to the weaknesses section above.

**Limitations:**

Yes the limitations are stated in the main paper.

---

> ### Author Rebuttal · Authors · 2024-08-05
>
> Thanks for the comments. All responses below will be put into revision.
>
> **[W1: Notion unclear]**
>
> We will revise Figure 1 to include a more intuitive display with 3D effect that demonstrates how multiple layers of AlphaTablets are composed to render each pixel in the final image.
>
> **[W2: Detailed caption for figures]**
>
> We will revise the captions of figures with more details and comprehensive explanations.
>
> **[W3: Tablet Storage]**
>
> As a 3D planar representation, AlphaTablets are designed to be lightweight. For example, the demo scene requires only 20MB storage, including all the texture maps in a lossless PNG format. Similar to mesh, advanced techniques such as texture atlas and compression can also be applied to further improve the storage efficiency.
>
> Once again, we thank the reviewer for the valuable comments that helps improve our paper.

---

> > ### Comment · Reviewer_VGFx · 2024-08-13
> >
> > Thanks for the response. I will keep my initial rating.

---

### Author Rebuttal · Authors · 2024-08-05

Please see the attached PDF file for figures and captions.

---

### Decision · Program_Chairs · 2024-09-25

**Decision:**

Accept (poster)

**Comment:**

All the reviewers are positive and recommending to accept the submission. The AC agrees with their decision. Please read the weakness  comments and the reviewers' concerns carefully and prepare the camera ready.